# A Teacher-Student Framework for
# Zero-Resource Neural Machine Translation

## Abstract

While end-to-end neural machine translation (NMT) has made remarkable progress recently, it still suffers from the data scarcity problem for low resource language pairs and domains. In this paper, we propose a method for zero-resource NMT by assuming that parallel sentences have close probabilities of generating a sentence in a third language. Based on the assumption, we are able to train a source-to-target NMT model without parallel corpora available ("student") guided by an existing pivot-to-target NMT model ("teacher") on a source-pivot corpus. Experimental results show that the proposed method significantly improves over a baseline pivot-based model by +3.0 BLEU points across various language pairs.

## 1 Introduction

Neural machine translation (NMT) (Kalchbrenner and Blunsom, 2013; Sutskever et al., 2014; Bahdanau et al., 2014), which directly models the translation process in an end-to-end way, has attracted intensive attention from the community. Although NMT has achieved state-of-the-art translation performance on resource-rich language pairs such as English-French and German-English (Luong et al., 2015; Jean et al., 2015; Wu et al., 2016), it still suffers from the unavailability of large-scale parallel corpora for translating low resource languages. Due to the large parameter space, neural models usually learn poorly from low-count events, resulting in a poor choice for low resource language pairs. Zoph et al. (2016) indicate that NMT obtains much worse translation quality than a statistical machine translation (SMT) system on low-resource languages.

As a result, a number of authors have endeavored to explore methods for translating language pairs without parallel corpora available. These methods can be roughly divided into two broad categories: *multilingual* and *pivot-based*. Firat et al. (2016b) present a multi-way, multilingual model with shared attention to achieve zero-resource translation. They fine-tune the attention part using pseudo bilingual sentences for the zero-resource language pair. Another direction is to develop a universal NMT model in multilingual scenarios (Johnson et al., 2016; Ha et al., 2016). They use parallel corpora of multiple languages to train one single model, which is then able to translate a language pair without parallel corpora available. Although these approaches prove to be effective, the combination of multiple languages in modeling and training leads to significantly increased complexity.

Another direction is to achieve source-to-target NMT without parallel data via a pivot, which is either text (Cheng et al., 2016) or image (Nakayama and Nishida, 2016b). Cheng et al. (2016) propose a pivot-based method to zero-resource NMT: it first translates the source language to a pivot language, which is then translated to the target language. Nakayama and Nishida (2016b) show that using multimedia information as pivot also significantly benefit zero-resource translation. However, pivot-based approaches usually need to divide the decoding process into two steps, which is not only more computationally expensive, but also potentially suffers from the error propagation problem (Zhu et al., 2013).

In this paper, we propose a new method for zero-resource neural machine translation. Our method assumes that parallel sentences should have close probabilities of generating a sentence in a third language. To train a source-to-target NMT

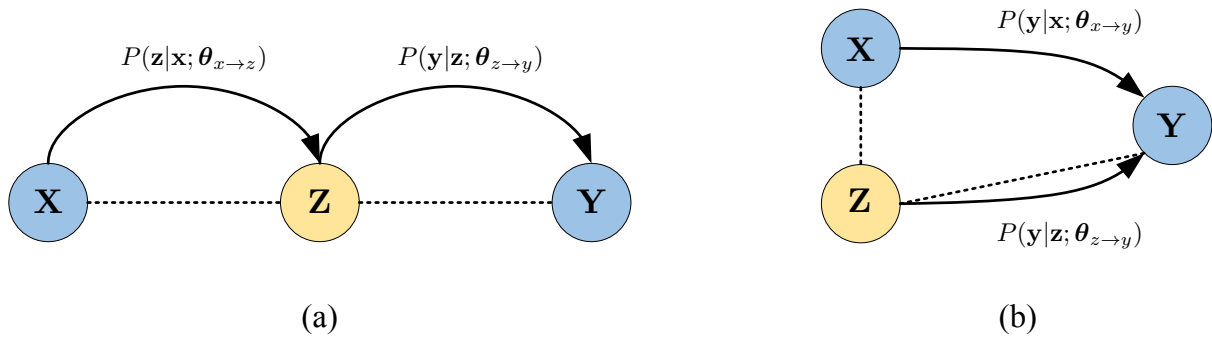

Figure 1: (a) The pivot-based approach and (b) the teacher-student approach to zero-resource neural machine translation. $\mathbf{X}$, $\mathbf{Y}$, and $\mathbf{Z}$ denote source, target, and pivot languages, respectively. We use a dashed line to denote that there is a parallel corpus available for the connected language pair. Solid lines with arrows represent translation directions. The pivot-based approach leverages a pivot to achieve indirect source-to-target translation: it first translates $\mathbf{x}$ into $\mathbf{z}$, which is then translated into $\mathbf{y}$. Our approach directly trains the intended source-to-target model ("student") on a source-pivot parallel corpus, with the guidance of an existing pivot-to-target model ("teacher"). Our training algorithm is based on the translation equivalence assumption: if $\mathbf{x}$ is a translation of $\mathbf{z}$, then $P(\mathbf{y}|\mathbf{x}; \boldsymbol{\theta}_{x \to y})$ should be close to $P(\mathbf{y}|\mathbf{z}; \boldsymbol{\theta}_{z \to y})$.

model without parallel corpora ("student"), we leverage an existing pivot-to-target NMT model ("teacher") to guide the learning process of the student model on a source-pivot parallel corpus. As compared with the pivot-based approaches (Cheng et al., 2016; Nakayama and Nishida, 2016b), our method allows direct modeling of the intended NMT model, without the need to divide training or decoding into two steps. This strategy not only improves efficiency but also avoids error propagation in decoding. Experiments on the Europarl and WMT datasets show that our approach achieves significant improvements in terms of both translation quality and decoding efficiency over a baseline pivot-based approach to zero-resource NMT on Spanish-French and German-French translation tasks.

## 2 Background

Neural machine translation (Sutskever et al., 2014; Bahdanau et al., 2014) advocates the use of neural networks to model the translation process in an end-to-end manner. As a data-driven approach, NMT treats parallel corpora as the major source for acquiring translation knowledge.

Let $\mathbf{x}$ be a source-language sentence and $\mathbf{y}$ be a target-language sentence. We use $P(\mathbf{y}|\mathbf{x}; \boldsymbol{\theta}_{x \to y})$ to denote a source-to-target neural translation model, where $\boldsymbol{\theta}_{x \to y}$ is a set of model parame-

ters. Given a source-target parallel corpus $D_{x,y}$, which is a set of parallel source-target sentences, the model parameters can be learned by maximizing the log-likelihood of the parallel corpus:

$$\hat{\boldsymbol{\theta}}_{x \to y} = \underset{\boldsymbol{\theta}_{x \to y}}{\operatorname{argmax}} \left\{ \sum_{\langle \mathbf{x}, \mathbf{y} \rangle \in D_{x,y}} \log P(\mathbf{y}|\mathbf{x}; \boldsymbol{\theta}_{x \to y}) \right\}$$

Given learned model parameters $\hat{\boldsymbol{\theta}}_{x \to y}$, the decision rule for finding the translation with the highest probability for a source sentence $\mathbf{x}$ is given by

$$\hat{\mathbf{y}} = \underset{\mathbf{y}}{\operatorname{argmax}} \left\{ P(\mathbf{y}|\mathbf{x}; \hat{\boldsymbol{\theta}}_{x \to y}) \right\} \quad (1)$$

As a data-driven approach, NMT heavily relies on the availability of large-scale parallel corpora to deliver state-of-the-art translation performance (Wu et al., 2016). Zoph et al. (2016) report that NMT obtains much lower BLEU scores than SMT if only small-scale parallel corpora are available. The heavy dependence on the quantity of training data poses a severe challenge for NMT to translation zero-resource language pairs.

Simple and easy-to-implement, pivot-based methods have been widely used in SMT for translating zero-resource language pairs (De Gispert and Marino, 2006; Cohn and Lapata, 2007; Utiyama and Isahara, 2007; Wu and Wang, 2007;

Bertoldi et al., 2008; Wu and Wang, 2009; Za-habi et al., 2013; Kholy et al., 2013). As pivot-based methods are agnostic to model structures, they have been adapted to NMT recently (Cheng et al., 2016; Nakayama and Nishida, 2016b).

Figure 1 illustrates the basic idea of pivot-based approaches to zero-resource NMT (Cheng et al., 2016; Nakayama and Nishida, 2016b). Let $\mathbf{X}$, $\mathbf{Y}$, and $\mathbf{Z}$ to denote source, target, and pivot languages. We use dashed lines to denote language pairs with parallel corpora available and solid lines with arrows to denote translation directions.

Intuitively, the source-to-target translation can be indirectly modeled by bridging two NMT models via a pivot:

$$P(\mathbf{y}|\mathbf{x}; \boldsymbol{\theta}_{x\to z}, \boldsymbol{\theta}_{z\to y})$$
$$= \sum_{\mathbf{z}} P(\mathbf{z}|\mathbf{x}; \boldsymbol{\theta}_{x\to z}) P(\mathbf{y}|\mathbf{z}; \boldsymbol{\theta}_{z\to y}) \quad (2)$$

As shown in Figure 1, the pivot-based approaches assume that the source-to-target parallel corpus $D_{x,z}$ and the pivot-to-target parallel corpus $D_{z,y}$ are available. As it is impractical to enumerate all possible pivot sentences, the two NMT models are trained separately in practice:

$$\hat{\boldsymbol{\theta}}_{x\to z} = \underset{\theta_{x\to z}}{\operatorname{argmax}} \left\{ \sum_{\langle \mathbf{x}, \mathbf{z} \rangle \in D_{x,z}} \log P(\mathbf{z}|\mathbf{x}; \boldsymbol{\theta}_{x\to z}) \right\}$$

$$\hat{\boldsymbol{\theta}}_{z\to y} = \underset{\theta_{z\to y}}{\operatorname{argmax}} \left\{ \sum_{\langle \mathbf{z}, \mathbf{y} \rangle \in D_{z,y}} \log P(\mathbf{y}|\mathbf{z}; \boldsymbol{\theta}_{z\to y}) \right\}$$

Due to the exponential search space of pivot sentences, the decoding process of translating an unseen source sentence $\mathbf{x}$ has to be divided into two steps:

$$\hat{\mathbf{z}} = \underset{\mathbf{z}}{\operatorname{argmax}} \left\{ P(\mathbf{z}|\mathbf{x}; \hat{\boldsymbol{\theta}}_{x\to z}) \right\} \quad (3)$$

$$\hat{\mathbf{y}} = \underset{\mathbf{y}}{\operatorname{argmax}} \left\{ P(\mathbf{y}|\hat{\mathbf{z}}; \hat{\boldsymbol{\theta}}_{z\to y}) \right\} \quad (4)$$

Despite its simplicity, the above two-step decoding process potentially suffers from the error propagation problem (Zhu et al., 2013): the translation errors made in the first step (i.e., source-to-pivot translation) will affect the second step (i.e., pivot-to-target translation).

Therefore, it is necessary to explore methods for direct modeling of source-to-target translation without parallel corpora available.

## 3 Approach

### 3.1 Assumptions

In this work, we propose to directly model the intended source-to-target neural translation based on a teacher-student framework. The basic idea is to use a pre-trained pivot-to-target model ("teacher") to guide the learning process of a source-to-target model without training data available ("student") on a source-pivot parallel corpus.

As shown in Figure 1(b), we still assume that a source-to-pivot parallel corpus $D_{x,z}$ and a pivot-to-source parallel corpus $D_{z,y}$ are available. Unlike pivot-based approaches, we first use the pivot-to-target parallel corpus $D_{z,y}$ to obtain a **teacher model** $P(\mathbf{y}|\mathbf{z}; \hat{\boldsymbol{\theta}}_{z\to y})$, where $\hat{\boldsymbol{\theta}}_{z\to y}$ is a set of learned model parameters. Then, the teacher model "teaches" the **student model** $P(\mathbf{y}|\mathbf{x}; \boldsymbol{\theta}_{x\to y})$ based on the following assumption.

**Assumption 1** *If a source sentence $\mathbf{x}$ is a translation of a pivot sentence $\mathbf{z}$, then the probability of generating a target sentence $\mathbf{y}$ from $\mathbf{x}$ should be close to that from its counterpart $\mathbf{z}$.*

We can move a further step to introduce a word-level assumption:

**Assumption 2** *If a source word $x$ is a translation of a pivot word $z$, then the probability of generating a target sentence $y$ from $x$ should be close to that from its counterpart $z$.*

The two assumptions are empirically verified in our experiments (see Table 2). In the following subsections, we will introduce two approaches to zero-resource neural machine translation based on the two assumptions.

### 3.2 Sentence-Level Teaching

Given a source-pivot parallel corpus $D_{x,z}$, our training objective based on Assumption 1 is defined as follows:

$$\mathcal{J}_{\text{SENT}}(\boldsymbol{\theta}_{x\to y})$$
$$= \sum_{\langle \mathbf{x}, \mathbf{z} \rangle \in D_{x,z}} \text{KL}\Big(P(\mathbf{y}|\mathbf{z}; \hat{\boldsymbol{\theta}}_{z\to y}) \Big\| P(\mathbf{y}|\mathbf{x}; \boldsymbol{\theta}_{x\to y})\Big) \quad (5)$$

where the KL divergence sums over all possible target sentences:

$$\text{KL}\Big(P(\mathbf{y}|\mathbf{z}; \hat{\boldsymbol{\theta}}_{z\to y}) \Big\| P(\mathbf{y}|\mathbf{x}; \boldsymbol{\theta}_{x\to y})\Big)$$
$$= \sum_{\mathbf{y} \in \mathcal{Y}} P(\mathbf{y}|\mathbf{z}; \hat{\boldsymbol{\theta}}_{z\to y}) \log \frac{P(\mathbf{y}|\mathbf{z}; \hat{\boldsymbol{\theta}}_{z\to y})}{P(\mathbf{y}|\mathbf{x}; \boldsymbol{\theta}_{x\to y})} \quad (6)$$

As the teacher model parameters are fixed, the training objective can be equivalently written as

$$
\begin{aligned}
&\mathcal{J}_{\text{SENT}}(\boldsymbol{\theta}_{x \to y}) \\
&= - \sum_{\langle \mathbf{x}, \mathbf{z} \rangle \in D_{x,z}} \mathbb{E}_{\mathbf{y}|\mathbf{z}; \hat{\boldsymbol{\theta}}_{z \to y}} \Big[ \log P(\mathbf{y}|\mathbf{x}; \boldsymbol{\theta}_{x \to y}) \Big] \quad (7)
\end{aligned}
$$

In training, our goal is to find a set of source-to-target model parameters that maximizes the training objective:

$$
\hat{\boldsymbol{\theta}}_{x \to y} = \underset{\boldsymbol{\theta}_{x \to y}}{\operatorname{argmax}} \Big\{ \mathcal{J}_{\text{SENT}}(\boldsymbol{\theta}_{x \to y}) \Big\} \quad (8)
$$

However, a major difficulty is the intractability in calculating the gradients because of the exponential search space of target sentences. To address this problem, it is possible to construct a subspace by either sampling (Shen et al., 2016), generating a $k$-best list (Yong et al., 2016) or mode approximation (Kim and Rush, 2016). Then, standard stochastic gradient descent algorithms can be used to optimize model parameters.

### 3.3 Word-Level Teaching

Instead of minimizing the KL divergence between the teacher and student models at the sentence level, we further define a training objective based on Assumption 2 at the word level:

$$
\begin{aligned}
&\mathcal{J}_{\text{WORD}}(\boldsymbol{\theta}_{x \to y}) \\
&= \sum_{\langle \mathbf{x}, \mathbf{z} \rangle \in D_{x,z}} \mathbb{E}_{\mathbf{y}|\mathbf{z}; \hat{\boldsymbol{\theta}}_{z \to y}} \Big[ J(\mathbf{x}, \mathbf{y}, \mathbf{z}, \hat{\boldsymbol{\theta}}_{z \to y}, \boldsymbol{\theta}_{x \to y}) \Big] \quad (9)
\end{aligned}
$$

where

$$
\begin{aligned}
&J(\mathbf{x}, \mathbf{y}, \mathbf{z}, \hat{\boldsymbol{\theta}}_{z \to y}, \boldsymbol{\theta}_{x \to y}) \\
&= \sum_{j=1}^{|\mathbf{y}|} \text{KL}\Big( P(y|\mathbf{z}, \mathbf{y}_{<j}; \hat{\boldsymbol{\theta}}_{z \to y}) \Big\| \\
&\qquad\qquad P(y|\mathbf{x}, \mathbf{y}_{<j}; \boldsymbol{\theta}_{x \to y}) \Big) \quad (10)
\end{aligned}
$$

Equation (9) suggests that the teacher model $P(\mathbf{y}|\mathbf{z}, \mathbf{y}_{<j}; \hat{\boldsymbol{\theta}}_{z \to y})$ "teaches" the student model $P(\mathbf{y}|\mathbf{x}, \mathbf{y}_{<j}; \boldsymbol{\theta}_{x \to y})$ in a word-by-word way. Note that the KL-divergence between two models is defined at the word level:

$$
\begin{aligned}
&\text{KL}\Big( P(y|\mathbf{z}, \mathbf{y}_{<j}; \hat{\boldsymbol{\theta}}_{z \to y}) \Big\| P(y|\mathbf{x}, \mathbf{y}_{<j}; \boldsymbol{\theta}_{x \to y}) \Big) \\
&= \sum_{y \in \mathcal{V}_y} P(y|\mathbf{z}, \mathbf{y}_{<j}; \hat{\boldsymbol{\theta}}_{z \to y}) \log \frac{P(y|\mathbf{z}, \mathbf{y}_{<j}; \hat{\boldsymbol{\theta}}_{z \to y})}{P(y|\mathbf{x}, \mathbf{y}_{<j}; \boldsymbol{\theta}_{x \to y})}
\end{aligned}
$$

| Corpus | Direction | Train | Dev. | Test |
|--------|-----------|-------|------|------|
| Europarl | Es→ En | 850K | 2,000 | 2,000 |
| | De→ En | 840K | 2,000 | 2,000 |
| | En→ Fr | 900K | 2,000 | 2,000 |
| WMT | Es→ En | 6.78M | 3,003 | 3,003 |
| | En→ Fr | 9.29M | 3,003 | 3,003 |

Table 1: Data statistics. For the Europarl corpus, we evaluate our approach on Spanish-French (Es-Fr) and Germany-French (De-Fr) translation tasks. For the WMT corpus, we our approach on the Spanish-French (Es-Fr) translation task. English is used a pivot language in all experiments.

Therefore, our goal is to find a set of source-to-target model parameters that maximizes the training objective:

$$
\hat{\boldsymbol{\theta}}_{x \to y} = \underset{\boldsymbol{\theta}_{x \to y}}{\operatorname{argmax}} \Big\{ \mathcal{J}_{\text{WORD}}(\boldsymbol{\theta}_{x \to y}) \Big\} \quad (11)
$$

We use similar approach for approximating the full search space with sentence-level teaching.

## 4 Experiments

### 4.1 Setup

We evaluate our approach on the Europarl and WMT corpora. To compare with the pivot-based methods, we use the same dataset as in the paper (Cheng et al., 2016). The evaluation metric is case-insensitive BLEU (Papineni et al., 2002) as calculated by the *multi-bleu.perl* script. All the experiments treat English as the pivot language and French as the target language.

For the Europarl corpus (Koehn, 2005), we evaluate our proposed model on Spanish-French (Es-Fr) and Germany-French (De-Fr) translation tasks in a zero-resource scenario. To build non-overlapping source-to-pivot and pivot-to-target datasets, we split pivot sentences shared by the original source-to-pivot and pivot-to-target corpora into two equal parts and merge them separately to the remaining source-to-pivot and pivot-to-target corpora. Both the development and test datasets are from shared task 2006. All the sentences are tokenized by the *tokenize.perl* script. To deal with out-of-vocabulary words, we also adopt byte pair encoding (BPE) (Sennrich et al., 2016) to split words into sub-words. The size of sub-words is set to 30K for each language.

For the WMT corpus, we evaluate our approach on a Spanish-French (Es-Fr) translation task with

| | Approx. | $P(\mathbf{y}|\mathbf{x}; \boldsymbol{\theta}_{x \to y})$ | |
|---|---|---|---|
| | | random init. | trained |
| $\mathcal{J}_{\text{SENT}}$ | greedy | 313.0 | 53.5 |
| | beam | 323.5 | 51.8 |
| $\mathcal{J}_{\text{WORD}}$ | greedy | 274.0 | 37.1 |
| | beam | 288.7 | 36.5 |
| | sampling | 268.6 | 40.5 |

Table 2: Approximate KL divergence from source-to-target to pivot-to-target model. We use a random initialized source-to-target model for comparison.

a zero-resource setting. We combine the following corpora to form the Es-En and En-Fr parallel corpora: Common Crawl, News Commentary, Europarl v7 and UN. All the sentences are tokenized by the *tokenize.perl* script. Newstest2011 serves as the development set and newstest2012 and newstest2013 serve as test sets. We use case-sensitive BLEU to evaluate translation results. BPE is also used to reduce the vocabulary size. The size of sub-words is set to 43K, 33K, 43K for Spanish, English and French, respectively. See Table 1 for detailed statistics for both Europarl and WMT corpora.

We leverage a tutorial NMT open-source code implemented by Theano for all the experiments. [1] and compare our approach with state-of-the-art multilingual method (Firat et al., 2016b) and pivot-based method (Cheng et al., 2016) with the following variations:

1. Sentence-Level Teaching: for simplicity, we use mode as suggested in (Kim and Rush, 2016) to approximate the target sentence space, which is the result of running beam search on the pivot sentences and taking the highest-scoring target sentence with the teacher model. Beam size with K = 1 (greedy decoding) and K = 5 are investigated in our paper, denoted as *sent-greedy* and *sent-beam*, respectively.

2. Word-Level Teaching: we use the same mode approximation approach as in sentence-level teaching, denoted as *word-greedy* (beam search with K=1) and *word-beam* (beam search with K=5) respectively. Besides, Monte Carlo estimation by sampling from the

---

[1] *dl4mt-tutorial*: https://github.com/nyu-dl

teacher model is also investigated since it introduces more diverse data, denoted as *word-sampling*.

## 4.2 Assumptions Verification

To verify the assumptions in Section 3.1, we train a source-to-target translation model $P(\mathbf{y}|\mathbf{x}; \boldsymbol{\theta}_{x \to y})$ and a pivot-to-target translation model $P(\mathbf{y}|\mathbf{z}; \boldsymbol{\theta}_{z \to y})$ using trilingual corpus. Then we use these two models to calculate $\mathcal{J}_{\text{SENT}}$ and $\mathcal{J}_{\text{WORD}}$, which reveals how close the sentence-level and word-level distributions are, respectively. Since it is difficult to measure $\mathcal{J}_{\text{SENT}}$ and $\mathcal{J}_{\text{WORD}}$ directly , we instead calculate their approximations as illustrated in Section 4.1. We report average KL divergence with different approximations evaluated on the aligned trilingual development set $\mathcal{E}_{xzy}$ with 2000 sentences from shared task 2006 as shown in Table 1. We also report KL Divergence from a randomly initialized NMT model $P_{random}$ to pivot-to-target $P(\mathbf{y}|\mathbf{z}, \boldsymbol{\theta}_{z \to y})$ for comparison.

The results agree with our assumptions since significantly smaller KL divergence is observed for $\text{KL}\big(P(\mathbf{y}|\mathbf{z}; \boldsymbol{\theta}_{z \to y})||P(\mathbf{y}|\mathbf{x}; \boldsymbol{\theta}_{x \to y})\big)$. For sentence-level distributions, smaller KL divergence for beam search approximation is observed compared with greedy approximation, indicating that mode produced by the beam search results of $P(\mathbf{y}|\mathbf{z}; \boldsymbol{\theta}_{z \to y})$ have higher probabilities in $P(\mathbf{y}|\mathbf{x}; \boldsymbol{\theta}_{x \to y})$ compared with the greedy results. Thus, we suspect that the sent-beam method will surpass the sent-greedy method. For word-level results, KL divergence of sampling approximation is the highest, indicating poorer performance than the other two methods. However, sampling methods introduce more data diversity at the target side. Thus it is harder to decide which factor dominates the training process. We leave further discussion to Section 4.3.

## 4.3 Results on the Europarl Corpus

### 4.3.1 Comparison with pivot-based methods

Table 3 gives BLEU scores on the Europarl corpus of our best performing sentence-level method (sent-beam) and word-level method (word-sampling) compared with pivot-based methods (Cheng et al., 2016). The same data preprocessing are applied in (Cheng et al., 2016) as in our experiments. We find that our sentence-level method and word-level method outperform all of their zero-resource approaches across language

| Method | | BLEU | |
|---|---|---|---|
| | | Es→ Fr | De→ Fr |
| Cheng.# | pivot | 29.79 | 23.70 |
| | hard | 29.93 | 23.88 |
| | soft | 30.57 | 23.79 |
| | likelihood$^\dagger$ | 32.59 | 25.93 |
| Ours | sent-beam | 31.64 | 24.39 |
| | word-sampling | 33.86 | 27.03 |

Table 3: Comparison with previous work on Spanish-French and German-French translation tasks from Europarl corpus. English is treated as the pivot language. † denotes using parallel source-to-target corpus. # denotes methods in Cheng et al. (2016).

pairs. Our word-sampling method improves their best zero-resource results on Spanish-French translation by 3.29 BLEU and Germany-French translation by 3.15 BLEU. Besides that, our word-sampling method surprisingly obtains improvement over the likelihood method, which leverages source-to-target parallel corpus and also takes much longer to train. The significant improvements can be explained by the error propagation problem of pivot-based methods that translation error of the source-pivot translation system will be transferred to the pivot-target translation.

### 4.3.2 Comparison of Sentence-Level and Word-Level Methods

Table 4 shows BLEU scores on the Europarl corpus of our proposed five methods.

For sentence-level approaches, sent-beam outperforms sent-greedy by 0.59 BLEU over es→ fr translation and 2.51 BLEU over de→ fr translation. The results are in line with our observations in Section 4.2 that sentence-level KLD by beam approximation is smaller than that by greedy approximation. However, as the time complexity grows linearly in the number of beams $k$, the better performance is achieved at the expense of beam search time.

For word-level experiments, we observe that word-sampling performs much better than the other two methods: 1.94 BLEU on es→ fr translation and 2.65 BLEU on de→ fr translation; 1.88 BLEU on es→ fr translation and 2.84 BLEU on de→ fr translation. Althrough word-level KLD calculated by Monte Carlo estimation is larger

| Methods | es→ fr | | de→ fr | |
|---|---|---|---|---|
| | dev | test | dev | test |
| sent-greedy | 31.00 | 31.05 | 22.34 | 21.88 |
| sent-beam | 31.57 | 31.64 | 24.95 | 24.39 |
| word-greedy | 31.37 | 31.92 | 24.72 | 25.15 |
| word-beam | 30.81 | 31.21 | 24.64 | 24.19 |
| word-sampling | 33.65 | 33.86 | 26.99 | 27.03 |

Table 4: Comparison of our proposed methods on Spanish-French and German-French translation tasks from Europarl corpus. English is treated as the pivot language.

| Methods | #sents | | | BLEU |
|---|---|---|---|---|
| | de-en | en-fr | de-fr | |
| MLE | − | − | 840K | 26.06 |
| | - | - | 1.5M | 29.02 |
| Ours | 840K | 900K | - | 27.03 |

Table 5: Comparison on German-French translation tasks from Europarl corpus with stardard NMT.

than that by mode approximation, word-sampling introduces more data diversity for training, which dominates the effect of KLD difference.

### 4.3.3 BLEU Score over Iterations

Figure 3 shows valid loss [2] and BLEU score over iterations. We observe that word-level models tend to have lower valid loss compared with sentence-level methods. Generally, models with lower valid loss are inclined to have higher BLEU. Our results indicate that this is not necessarily the case: sent-beam converges to +0.82 the BLEU on validation set with -13 valid loss compared with the word-beam method. Kim and Rush (2016) claims a similar observation in data distillation for NMT and provides an explanation that student distributions are more peaked for sentence-level models. This is indeed the case in our result: on Germany-French translation task the argmax for the sent-beam student model (on average) accounts approximately for 3.49% of the total probability mass, while the corresponding number is 1.25% for the word-beam student model and 2.60% for the teacher model. However, argmax for the sentence-level model in (Kim and Rush, 2016) comprises around 15% output probability, while ours only accounts for 3.49%. It possibly reveals that it is more diffi-

---

[2] *Valid loss*: the average NLL of sentence pairs on the validation set.

| Method | # Training Sents | | # Para. | BLEU | |
|---|---|---|---|---|---|
| | Es→ En | En→ Fr | | Newstest2012 | Newstest2013 |
| *Existing zero-resource NMT systems* | | | | | |
| Cheng et al. (2016)[†] | pivot | 6.78M | 9.29M | - | 19.81 | - |
| Cheng et al. (2016)[†] | likelihood | 6.78M | 9.29M | 100K | 21.48 | - |
| Firat et al. (2016b) | one-to-one | 34.71M | 65.77M | - | 17.59 | 17.61 |
| Firat et al. (2016b)[†] | many-to-one | 34.71M | 65.77M | - | 21.33 | 21.19 |
| *Our zero-resource NMT systems* | | | | | |
| | word-sampling | 6.78M | 9.29M | - | 25.26 | 25.06 |

Table 6: Comparison with previous work on Spanish-French translation in zero resource scenario over WMT corpus. The BLEU scores are case sensitive. †: the pivot path involved in the translation path during testing.

cult to model around the teacher's mode for cross-task teaching than in data distillation.

### 4.3.4 Comparison with Standard NMT

Table 5 gives the comparison on German-French zero-resource translation between word-sampling approach and standard NMT model trained by MLE. Our proposed method even slightly outperforms standard NMT system trained with parallel source-to-target corpus when this corpus contains similar number of sentences as source-to-pivot corpus, demonstrating that the effectiveness of our methods to learn from teacher model. This phenomenon can be explained by that the total size of corpus involved in our method is twice that of standard MLE. The translation knowledge contained in other language pairs benefit source-to-target translation, as observed in multilingual scenario (Johnson et al., 2016; Ha et al., 2016).

### 4.3.5 BLEU Score with Low-Resource Source-to-Pivot Corpus

Our proposed method can also be applied to zero-resource NMT with low source-to-pivot corpus. Specifically, the size of source-pivot corpus is magnitude smaller than that of pivot-target corpus. This setting makes sense in applications. For example, there are significantly fewer Urdu-English corpus available than English-French corpus.

To fulfill this task, we combine our best performing word-sampling method with the initialization and parameter freezing strategy proposed in (Zoph et al., 2016). We set the size of De-En corpus to be $100K$ and use the same teacher model trained with $900K$ corpus. Table 7 gives the BLEU score of our method on De-Fr translation compared with three other methods. Noting that our task is much harder than (Zoph et al.,

| Methods | corpus | | | BLEU |
|---|---|---|---|---|
| | De-En | De-Fr | En-Fr | |
| MLE | × | √ | × | 19.30 |
| *transfer*[†] | × | √ | √ | 22.39 |
| pivot | √ | × | √ | 17.32 |
| ours | √ | × | √ | 22.95 |

Table 7: Comparison on German-French translation tasks from Europarl corpus with $100K$ German-English resource. English is regarded as the pivot language. † denotes the transfer learning method in (Zoph et al., 2016). $100K$ parallel German-French corpus are used for MLE and *transfer*.

2016) (*transfer*) since they are given parallel De-Fr corpus directly. Surprisingly, our method outperforms all other methods. We significantly improve the baseline pivot method by 5.63 BLEU and the state-of-the-art low resource method *transfer* by 0.56 BLEU. We leave further evaluation of this method for future work. [3]

### 4.4 Results on the WMT Corpus

Word-sampling method obtains the best performance in our five proposed approaches according to experiments on the Europarl corpus. To further investigate this approach, we conduct experiment on large scale WMT corpus for es→ fr translation. Table 6 shows the results of our word-sampling method in comparison with other state-of-the-art baselines. Cheng et al.(2016) use the same training&development sets and the same

---

[3]We test initialization trick in our original settings and observe that with initialization, the BLEU starts to increase very fast (21.25 Bleu with 10000 iterations while without initialization, the number is 2.5). However the model converges to similar point as without initialization.

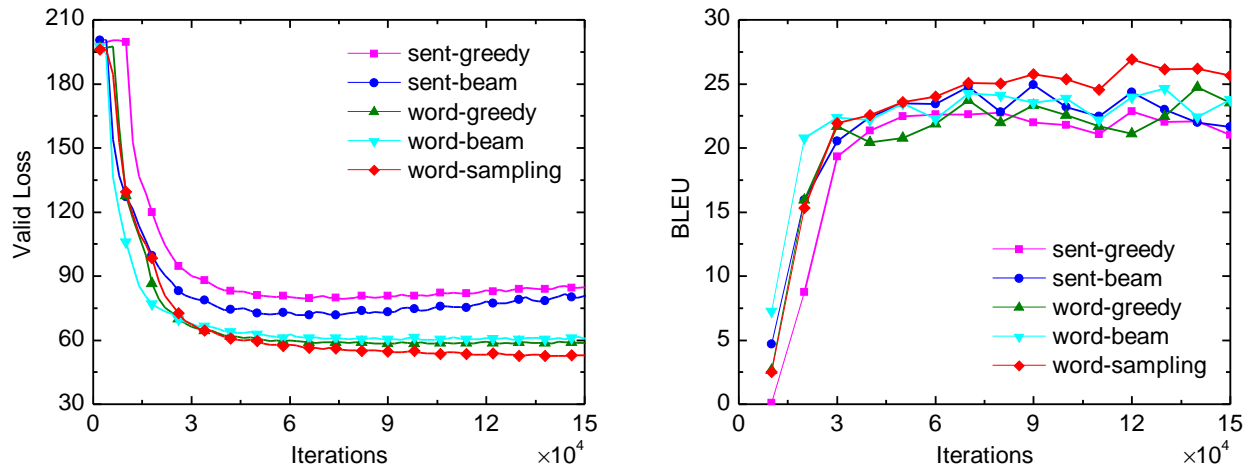

Figure 2: Validation loss and BLEU across iterations of our proposed methods.

preprocessing as ours. Firat et al.(2016b) uses a much larger training sets. Our method obtains significant improvement over the pivot baseline by 5.45 BLEU on *newstest2012* and over many-to-one 3.87 BLEU on *newstest2013*, not to mention that both method are depending on source-pivot-target decoding path.

## 5 Related Work

Training NMT system in zero-resource scenario by leveraging some other languages has attracted intensive attention in last year. Firat et al. (2016b) propose an approach which genuinely delivers the multi-way, multilingual NMT model proposed by (Firat et al., 2016a) to zero-resource translation. They use the multi-way NMT model trained by other language pairs to generate pseudo parallel corpus and fine-tune the attention mechanism of the multi-way NMT model to enable zero-resource translation. Several authors propose a universal encoder-decoder network in multilingual scenarios to perform zero-shot learning. This universal model extracts translation knowledge from multiple different languages, making zero-resource translation feasible without directly training (Johnson et al., 2016; Ha et al., 2016).

Besides multilingual NMT, another import line of research is pivot-based methods. Inspired by the widely used pivot idea in SMT (De Gispert and Marino, 2006; Cohn and Lapata, 2007; Utiyama and Isahara, 2007; Wu and Wang, 2007; Bertoldi et al., 2008; Zahabi et al., 2013; Kholy et al., 2013), Cheng et al.(2016) propose pivot-based NMT by simultaneously improving source

to pivot and pivot to target translation quality in order to improve source to target translation quality. Nakayama and Nishida (2016a) achieve zero-resource machine translation by utilizing image as pivot.

In (Kim and Rush, 2016), the author first introduce knowledge distilling in neural machine translation. They suggest to generate pseudo corpus to train the student network. Our work is highly related to theirs. However, we focus on zero-resource learning instead of model compression.

## 6 Conclusion

In this paper, we propose a novel framework to transfer the knowledge of the teacher model trained with rich-resource language pair into the student model with zero-resource with the help of source-to-pivot corpus. We introduce sentence-level and word-level teaching to guide the learning process of the student model. Our experiments on Europarl Corpus and WMT corpus across languages show that our proposed word-level sampling method can significantly outperforms the state-of-the-art pivot-based methods and multilingual methods in terms of both translation quality and decoding efficiency.

We also analyze zero-resource translation with low source-to-pivot corpus, and combine our word-level sampling method with initialization and parameter freezing suggested by (Zoph et al., 2016). The experiments on Europarl Corpus show that our approach obtains an significant improvement over pivot method significantly.

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
