# Peer review of "A Teacher-Student Framework for Zero-Resource Neural Machine Translation"

_ACL 2017 — decision unknown_

[Official Review · Reviewer 1 · rating 4 · confidence 4]
soundness 4 · originality 4 · clarity 5 · impact 3 · substance 4 · appropriateness 5 · meaningful comparison 4 · presentation format Oral Presentation

This paper proposes a novel strategy for zero-resource translation where
(source, pivot) and (pivot, target) parallel corpora are available. A teacher
model for p(target|pivot) is first trained on the (pivot, target) corpus, then
a student model for p(target|source) is trained to minimize relative entropy
with respect to the teacher on the (source, pivot) corpus. When using
word-level relative entropy over samples from the teacher, this approach is
shown to outperform previous variants on standard pivoting, as well as other
zero-resource strategies.

This is a good contribution: a novel idea, clearly explained, and with
convincing empirical support. Unlike some previous work, it makes fairly
minimal assumptions about the nature of the NMT systems involved, and hence
should be widely applicable.

I have only a few suggestions for further experiments. First, it would be
interesting to see how robust this approach is to more dissimilar source and
pivot languages, where intuitively the true p(target|source) and
p(target|pivot) will be further apart. Second, given the success of introducing
word-based diversity, it was surprising not to see a sentence n-best or
sentence-sampling experiment. This would be more costly, but not much more so
since you’re already doing beam search with the teacher. Finally, related to
the previous, it might be interesting to explore transition from word-based
diversity to sentence-based as the student converges and no longer needs the
signal from low-probability words.

Some further comments:

line 241: Despite its simplicity -> Due to its simplicity

277: target sentence y -> target word y

442: I assume that K=1 and K=5 mean that you compare probabilities of the most
probable and 5 most probable words in the current context. If so, how is the
current context determined - greedily or with a beam?

Section 4.2. The comparison with an essentially uniform distribution doesn’t
seem very informative here: it would be extremely surprising if p(y|z) were not
significantly closer to p(y|x) than to uniform. It would be more interesting to
know to what extent p(y|z) still provides a useful signal as p(y|x) gets
better. This would be easy to measure by comparing p(y|z) to models for p(y|x)
trained on different amounts of data or for different numbers of iterations.
Another useful thing to explore in this section would be the effect of the mode
approximation compared to n-best for sentence-level scores.

555: It’s odd that word beam does worse than word greedy, since word beam
should be closer to word sampling. Do you have an explanation for this?

582: The claimed advantage of sent-beam here looks like it may just be noise,
given the high variance of these curves.

[Official Review · Reviewer 2 · rating 3 · confidence 3]
soundness 3 · originality 3 · clarity 2 · impact 3 · substance 2 · appropriateness 5 · meaningful comparison 4 · presentation format Poster

In this paper the authors present a method for training a zero-resource NMT
system by using training data from a pivot language. Unlike other approaches
(mostly inspired in SMT), the author’s approach doesn’t do two-step
decoding. Instead, they use a teacher/student framework, where the teacher
network is trained using the pivot-target language pairs, and the student
network is trained using the source-pivot data and the teacher network
predictions of the target language.

- Strengths:

The results the authors present, show that their idea is promising. Also, the
authors present several sets of results that validate their assumptions.

- Weaknesses:

However, there are many points that need to be address before this paper is
ready for publication.

1)            Crucial information is missing

Can you flesh out more clearly how training and decoding happen in your
training framework? I found out that the equations do not completely describe
the approach. It might be useful to use a couple of examples to make your
approach clearer.

Also, how is the montecarlo sampling done? 

2)            Organization
The paper is not very well organized. For example, results are broken into
several subsections, while they’d better be presented together.  The
organization of the tables is very confusing. Table 7 is referred before table
6. This made it difficult to read the results.

3)            Inconclusive results:
After reading the results section, it’s difficult to draw conclusions when,
as the authors point out in their comparisons, this can be explained by the
total size of the corpus involved in their methods (621  ). 

4)            Not so useful information:
While I appreciate the fleshing out of the assumptions, I find that dedicating
a whole section of the paper plus experimental results is a lot of space. 

- General Discussion:

Other:
578:  We observe that word-level models tend to have lower valid loss compared
with sentence- level methods….
Is it valid to compare the loss from two different loss functions?

Sec 3.2, the notations are not clear. What does script(Y) means?
How do we get p(y|x)? this is never explained

Eq 7 deserves some explanation, or better removed.
320: What approach did you use? You should talk about that here
392 : Do you mean 2016?

Nitty-gritty:

742  : import => important
772  : inline citation style
778: can significantly outperform 
275: Assumption 2 needs to be rewritten … a target sentence y from x should
be close to that from its counterpart z.

[Official Review · Reviewer 3 · rating 4 · confidence 5]
soundness 4 · originality 4 · clarity 3 · impact 3 · substance 3 · appropriateness 5 · meaningful comparison 4 · presentation format Oral Presentation

- Strengths:
This is  a well written paper.
The paper is very clear for the most part.
The experimental comparisons are very well done.
The experiments are well designed and executed.
The idea of using KD for zero-resource NMT is impressive.

- Weaknesses:
There were many sentences in the abstract and in other places in the paper
where the authors stuff too much information into a single sentence. This could
be avoided. One can always use an extra sentence to be more clear.
There could have been a section where the actual method used could be explained
in a more detailed. This explanation is glossed over in the paper. It's
non-trivial to guess the idea from reading the sections alone.
During test time, you need the source-pivot corpus as well. This is a major
disadvantage of this approach. This is played down - in fact it's not mentioned
at all. I could strongly encourage the authors to mention this and comment on
it. 

- General Discussion:

This paper uses knowledge distillation to improve zero-resource translation.
The techniques used in this paper are very similar to the one proposed in Yoon
Kim et. al. The innovative part is that they use it for doing zero-resource
translation. They compare against other prominent works in the field. Their
approach also eliminates the need to do double decoding.

Detailed comments:
- Line 21-27 - the authors could have avoided this complicated structure for
two simple sentences.
Line 41 - Johnson et. al has SOTA on English-French and German-English.
Line 77-79 there is no evidence provided as to why combination of multiple
languages increases complexity. Please retract this statement or provide more
evidence. Evidence in literature seems to suggest the opposite.

Line 416-420 - The two lines here are repeated again. They were first mentioned
in the previous paragraph.
Line 577 - Figure 2 not 3!